# Vestibular Impairment in Patients with Vestibular Schwannoma: A Journey through the Pitfalls of Current Literature

**Davide Pisani** [1], **Federico Maria Gioacchini** [2], **Giuseppe Chiarella** [1,*], **Alessia Astorina** [1], **Filippo Ricciardiello** [3], **Alfonso Scarpa** [4], **Massimo Re** [2] **and Pasquale Viola** [1]

1    Unit of Audiology, Regional Centre of Cochlear Implants and ENT Diseases, Department of Experimental and Clinical Medicine, Magna Graecia University, 88100 Catanzaro, Italy
2    Ear, Nose, and Throat Unit, Department of Clinical and Molecular Sciences, Polytechnic University of Marche, Via Conca 71, 60020 Ancona, Italy
3    ENT Department, AORN Cardarelli, 80131 Napoli, Italy
4    Department of Medicine and Surgery, University of Salerno, 84084 Salerno, Italy
*    Correspondence: chiarella@unicz.it; Tel.: +39-096-1364-7124

**Abstract:** Vestibular Schwannoma is the most common tumour of Ponto Cerebellar Angle and is capable of strongly impacting the patient's quality of life. In recent decades, the proposals for the management of the disease have multiplied, just as the diagnostic capacity has improved. While in the past, the primary objective was the preservation of the facial function, and subsequently also of the auditory function, the attention to the vestibular symptomatology, which appears to be one of the main indicators of deterioration of quality of life, is still unsatisfactory. Many authors have tried to provide guidance on the best possible management strategy, but a universally recognized guideline is still lacking. This article offers an overview of the disease and the proposals which have advanced in the last twenty years, evaluating their qualities and defects in a critical reading.

**Keywords:** vestibular; schwannoma; dizziness; vertigo; compensation; rehabilitation; stereotactic; radiosurgery; microsurgery; quality of life





## 1. Introduction

Vestibular schwannoma (VS) is a slow-growing benign tumour arising from the Schwann Cells of the superior (SVN) or inferior (IVN) branch of the vestibulocochlear nerve [1], and the latter accounts for more than 90% of cases [2]. It is by far the most common tumour of the Cerebello–Pontine Angle (CPA) and represents 8–10% of all intracranial neoplasms [3]. The term "Vestibular Schwannoma" better describes the tumour's cell and nerve origin than the previously used "acoustic neuromas", which reflects how the hearing impairment is common and well described in patients suffering from VS [4].

The exact measurement of the incidence of VS is debated and lacking in certain data; however, it is subject to constant updates, considering the ever-growing incidental diagnoses deriving from the increased use of Magnetic Resonance Imaging (MRI) and its improved image resolution [5]. The estimated VS incidence is 1 in 100,000 per year, with a lifetime risk of 1 in 1000 [6,7]. A retrospective analysis of 46,414 MRI scans performed for reasons other than suspected VS identified eight, suggesting that undiagnosed VS may exist nearly in 0.02% of the population [8]. VS is associated with mutations or deletions of the nf2 gene on chromosome 22, which provide instructions to create a protein called merlin, also known as schwannomin, expressed in Schwann cells that wrap around and insulate nerves [9].

Merlin helps regulate important key points for cells' growth, shape, and adhesion to each other. Furthermore, it acts as a tumour suppressor. Disruption of this function is involved in tumorigenesis and metastasis [10]. The vast majority of VS is unilateral, equally affecting the right and left ear, and due to sporadic mutations, only 5% of VS are

associated with a dominant-inherited syndrome called Neurofibromatosis type 2 (NF2) syndrome [11]. Inactivation of the nf2 gene leads to the tumorigenesis of both NF2-sindromic and sporadic VS [12]. Bilateral VS is the trademark of the autosomal dominant inherited NF2 syndrome [13]. Usually, VS grow slowly, with an average growth rate of 0.4 to 2.9 mm/year [14,15]. Lesions can grow continuously or stop at a certain size, and even spontaneous shrinkage has been observed by Huang et al., as described in 3.8% of tumours following a watchful wait protocol [16]. VS patients suffer from dysfunction of structures anatomically close to the tumour. Typically, a VS originates from the internal (intracanalicular) acoustic meatus; if it extends outside the canal, the VS is classified as extracanalicular. The extracanalicular VS grows toward the pontocerebellar angle, leading it to compromise the function of nearby cranial nerves, brainstem nuclei or cerebellum. Thus, unilateral VS can present with a diverse spectrum of ipsilateral symptoms and signs [17]. Typical audiovestibular symptoms of VS are unilateral Hearing Loss (HL), tinnitus, vertigo, and instability. HL is a very common complaint in VS patients (95%), while tinnitus affects around 60%. Nearly half of all deaf patients experience tinnitus. Vertigo is reported in 28%, dizziness in 22% and disequilibrium and unsteadiness in nearly 40%. This may be also due to compression of the cerebellum, mostly by large extracanalicular tumours [18]. The HL leads to communication and social difficulties, as well as a tendency to isolate [19]. Tinnitus often accompanies sleep disturbances, fatigue, depression, concentration difficulties, alteration of the emotional sphere; vertigo and dizziness can cause anxiety, depression [20,21]. Patients complaining of HL usually undergo audiometry; objective vestibular evaluations are less frequent. Confirmation of unilateral deficit prompts further gadolinium-enhanced MRI of the internal acoustic meatus, which is the standard method for diagnosing VS. Diagnosis usually occurs in the fifth decade of life [6]. Given the way it grows, VS leave room for different management options, including microsurgery (MS), stereotactic radiosurgery (SRS), Gamma-knife radiosurgery (GK), fractionated radiotherapy (FRT) and watchful waiting/observation (OB). The choice of the therapeutic management method of a VS depends on size, symptoms and signs, the patient's age, the patient's health, as well as the patient's preference. Unfortunately, evidence suggests that patient choice is often biased by lack of information. This fits into a scenario in which the comfort of solid evidence-based guidelines is lacking [22]. From diagnosis onwards, the choice between longitudinal observation and intervention, the preference for one type of intervention rather than another, the follow-up interval, or the control methods of biometric or emotional indexes are not standardized, leaving room for interpretations that are not always coherent. In such a boundless horizon, most patient tends to rely on the expert's opinion, the lowest level of scientific evidence [23].

The purpose of our article is to provide a summary of the most representative literature of the last twenty years concerning the vestibular management of the VS, to provide thoughts on future developments, and to offer a quick access key to the vestibular aspects of the VS.

## 2. Materials and Methods

Two authors (D.P. and P.V.) independently selected studies for analysis according to the following inclusion criteria: VS patients with known preoperative data, treated with microsurgery, radiotherapy, or observational strategy; objective vestibular evaluation and/or subjective dizziness/vertigo/disequilibrium/gait/balance disorder assessment of outcomes; a proper post-treatment follow-up; type of study: retrospective or prospective study; articles in English. In November 2022, a PubMed literature search was performed matching "Vestibular Schwannoma" with "DHI", "vertigo", "dizziness", "balance", "quality of life" bound by the Boolean operator AND. The search produced a list of nearly 280 articles. Duplicates and anecdotal articles were eliminated. The results were restricted to the last 20 years. The exclusion criteria were as follows: had no original data and/or were editorial articles or conference abstracts; had no proper outcomes; papers with no full text available, Neurofibromatosis 1–2 or other histological neoplasm different from

Schwannoma, Schwannoma of the 7th cranial nerve; animal experiments, as well as cell-line studies or editorials and commentaries, and case reports. Two independent authors (D.P. and P.V.) screened the studies. The selected studies were identified by title, abstract and text in the first selection. The selections of the two authors were cross-referenced and any disagreements were discussed together with the third author (F.M.G.), then the complete text of the relevant studies was retrieved for validation (D.P., F.M.G.) before final inclusion in this review. We also attempted to identify articles not found in our initial PubMed query by checking the references of each article selected for inclusion. Any manuscript added was subjected to our inclusion and exclusion criteria according to the protocol already described. Each article was independently studied by two authors (D.P. and F.M.G.), who selected the salient points and weaknesses, validated by a subsequent discussion in the presence of a third author (P.V.).

## 3. Results

The main results of the critical reading of the selected articles are highlighted in Table 1 and shown below.

**Table 1.** P: prospective study; R: retrospective study; GK: gamma knife; SRS: Stereotactic radiosurgery; DHI: Dizziness Handicap Inventory; ABC: activities-specific balance confidence scale; HRT: head tilt response; VOG: caloric video-oculography; vHIT: video head impulse test; VOR: vestibulo-ocular reflex; SVV: subjective visual vertical; SF-36: short-form questionnaire SF-36; SHA: sinusoidal harmonic acceleration.

| Authors | Year | Study Design | Total n. of Patients | Operations Procedures (Type of Approach) | Radiosurgery Procedures | Strategies for Outcome's Measures | Authors' Reported Outcomes/Conclusions |
|---|---|---|---|---|---|---|---|
| Humphriss | 2003 | P | 100 | 100 (Translabyrinthine) | 0 | DHI | DHI scores becomes significantly worse between preoperative and 3-month postoperative time points but then does not continue to decline |
| Levo | 2004 | P/R | 177/44 | 166 RS + 11 TL/ 40 RS + 4 TL | 0 | posturography; questionnaire on gait and depression | Ageing is related to poor vestibular compensation. Visual feedback is ineffective in accidental slips and falls prevention, proper proprioceptive VRP improves postural stability and balance control |
| Enticott | 2005 | P | 65 | 65 (multiples) | 0 | DHI; SHA rotary tests | The study provided unique evidence that a program of simple vestibular exercises and education can speed the rate of compensation after vestibular schwannoma surgery |
| Pollock | 2006 | P | 82 | 36 (multiples) | 46 (GK) | DHI | The radiosurgical group had lower mean Dizziness Handicap Inventory scores (16.5 versus 8.4, $p = 0.02$) at last follow-up |
| Godefroy | 2007 | P | 17 | 17 (Translabyrinthine) | 0 | DHI | Translabyrinthine tumour removal significantly improved the patients' quality of life |
| Tuffarelli | 2007 | R | 386 | 386 (multiples) | 0 | DHI; ABC; questionnaire on oscillopsia | Disequilibrium influences handicap and disability after acoustic neuroma surgery. This symptom is also present after several years since surgery, and some patients perceived disequilibrium as disabling. |
| Park | 2011 | P | 59 | 0 | 59 (GK) | DHI | No significant decline in global QoL occurred after GK |
| Wagner | 2011 | P | 38 | 22 (Retrosigmoid) | 16 (SRS) | DHI; VOG | Loss of vestibular function was not strictly associated with a long-term deterioration of quality of life |

**Table 1.** *Cont.*

| Authors | Year | Study Design | Total n. of Patients | Operations Procedures (Type of Approach) | Radiosurgery Procedures | Strategies for Outcome's Measures | Authors' Reported Outcomes/Conclusions |
|---|---|---|---|---|---|---|---|
| Uehara | 2011 | P | 38 | 38 (Retrosigmoid) | 0 | DHI; posturography | The posturographic parameters and DHI scores at one week after surgery showed significant deterioration; the posturographic parameters and DHI scores for older patients tended to be worse than those for younger patients at 6 and 9 months after surgery |
| Breivik | 2013 | P | 237 | 113 (GK) + 124 (OB) | 113 | SF-36 | Symptom and QoL development did not differ significantly between the groups |
| Batuecas-Caletrio | 2013 | R | 49 | 49 (Multiples) | 0 | vHIT; DHI | Long-term follow-up after vestibular schwannoma surgery has shown that 22% of the patients display a particular abnormality in the VOR and these patients report the higher level of vestibular disability and handicap |
| Stavas | 2014 | P | 10 | 0 | 10 (SRS) | DHI | There were no statistically significant associations between radiation dose and change in DHI scores |
| Presutti | 2014 | R | 81 | 81 (Retrosigmoid) | 0 | SF-36 | Higher percentage of patients who did not complain of vertigo before surgery reported a worsening of QoL (57%) in comparison to subjects who had already experienced vertiginous attacks (26%) |
| Abboud | 2014 | P | 64 | 64 (Retrosigmoid) | 0 | VOG; rotational chair testing | Postoperative VOG demonstrated vestibular paresis in 80%; Rotary chair testing demonstrated normal or central compensation in 84% |
| Thomeer | 2015 | P | 48 | 48 (Transpetrosal) | 0 | VOG; DHI | Preoperatively, 77% experienced mild instability problems with a mean DHI score of 14.1 |
| Suarez | 2015 | R | 8 | 8 (n/a) | 0 | HRT | HTR test performed in a group of patients with chronic dizziness after acoustic neuroma surgery showed alterations in the gravitational vertical perception |
| Samii | 2017 | R | 19 | 19 (Retrosigmoid) | 0 | DHI | Compared with the control group, the DHI score at 3 weeks and 3 months after surgery was significantly worse. Vertigo was improved in all patients and completely resolved after 1 year in 17 patients. |
| Hrubà | 2019 | P | 52 | 52 (Retrosigmoid) | 0 | ABC; SVV; posturography | Significant improvement in SVV ($p < 0.05$), posturography parameters ($p < 0.05$) and ABC scores ($p < 0.05$) with postoperative rehabilitation program following surgery |
| Lee | 2019 | R | 115 | 0 | 115 (GK) | Clinical observation of vestibular symptoms | Thirtyseven (32%) patients developed vestibular symptoms within6 months post-GK; smaller vestibular schwannomas were significantlyassociated with higher rates of post-GK vestibular symptoms |
| Ermis | 2021 | R | 53 | 0 | 53 (SRS) | VOG | Patients with improved caloric function had received significantly lower mean ($1.5 \pm 0.7$ Gy, $p = 0.01$) and maximum doses ($4 \pm 1.5$ Gy, $p = 0.01$) to the vestibule. |

Humphriss et al. performed a prospective administration of Dizziness Handicap Inventory (DHI) [24] preoperatively and at 3 and 12 months after surgery in a sample of 100 consecutive patients treated with translabyrinthine (TL) excision of a unilateral sporadic VS [25]. The preoperative presentation of balance symptoms was recorded using a clinical evaluation and Electronystagmography (ENG). In total, 93 out of 100 patients were tested for vestibular impairment: 53 presented unsteadiness or disequilibrium, 8 had vertigo and 39 were asymptomatic. The incidence of preoperative disequilibrium was 53% and that of poor functional compensation status was 57%. The authors stated that DHI does not worsen postoperatively for most patients; when it does, it becomes significantly worse at the third-month follow-up, without further worsening. It has been observed that tumour size, sex and preoperative canal paresis affect the handicap score change, while age, central vestibular impairment and other symptoms have no effect.

In 2004, Levo et al. proposed a study on a retrospective sample of 177 patients who underwent surgery between 1979 and 1987 (166 RS and 11 TL) and on a prospective sample of 44 patients who underwent surgery between 1988 and 1991 (40 RS and 4 TL) [26]. In most cases, the intervention was radical, and the dimensions of the tumours varied between 4 mm and 60 mm in both samples, with a mean of about 20 mm. The sample of the retrospective study was analysed with posturographic examination and a questionnaire on gait and depression. The prospective sample was analysed posturographically, and rehabilitated as soon as possible with a postural training program. Patients in the retrospective study considered their gait to be normal in 69% of cases, but posturography corrected this figure at 32%, providing clear evidence of the impact of vestibular compensation on balance perception.

The authors underline how visual feedback alone is ineffective at preventing loss of balance and falls, while the enhancement of proprioceptive feedback through an appropriate Vestibular Rehabilitation Protocol (VRP) can be decisive in reducing risks, improving objective postural parameters and perception of balance in subjective tests (unfortunately, a standardized questionnaire is not used). In agreement with most authors, their analysis agrees on the harmful role of age on vestibular compensation abilities.

Enticott et al. studied the vestibular dysfunction in the first 12 weeks after surgery in a prospective investigation on a sample of 65 patients [27]. The authors stated that vestibular rehabilitation exercises did not modify the extent of the post-surgical functional deficit, as VOR gain in both exercise and control groups were not significantly different, but sped the rate of compensation after VS surgery. These results are in antithesis with those reported by Herdman et al. [28]. Such conflicting results are probably related to the different temporal planning of the study: Enticott started the postoperative vestibular evaluation only two weeks after the operation, while Herdman et al. ended the study within one week of the intervention. In our opinion, VOR benefits related to rehabilitation exercises were clear in the immediate postoperative stages.

We found an interesting study by Pollock et al. [29], which was conducted on a small sample of 82 patients (36 underwent surgical resection, 46 radiosurgery). Their first endpoint was preserving facial serviceability (they used a TL approach even in patients with serviceable hearing). Among the peculiar aspects of this study is the rigor with which the data were analysed by blinded investigators. This allowed the researchers to obtain two levels of evidence, despite some difficulties, such as, for example, the evaluation of the face by means of photographs, and although it was difficult to randomize the sample. Regarding the vestibular outcome after VS treatment, this study does not report any differences between microsurgery and irradiation, although it does report a better yield on the Health Status Questionnaire tests [30] (HSQ, a modified version of SF-36 [31]) for irradiated patients.

A small prospective study of 17 patients was presented by Godefroy et al. [32]. The authors analysed patients with LV and vestibular symptomatology by administering SF-36 and DHI questionnaires. While no differences were noted between the presurgical status and the status of patients 3 months postsurgery, a tangible improvement was noted at the

12-month follow-up. The authors underline how microsurgery is an effective strategy in patients with masses, even small ones, but has an impact on Quality of Life (QoL).

In 2007, Tufarelli et al. [33] presented a study on a sample of 386 consecutive patients undergoing VS surgery at a single centre. Patients were administered the DHI, Activities-specific Balance Confidence scale (ABC) [34] and one questionnaire for oscillopsia. Over 59% of patients did not perceive any disabling symptoms, while the remaining percentage indicated at least one. Oscillopsia had higher prevalence in females and in the middle cranial fossa approach. The authors found no significant differences for age, tumour size, surgical approach, or time interval from surgery, while they found statistical significance for total, emotional, and physical DHI scores. The ABC scores are significantly congruent with the DHI scores; most patients indicate a moderate handicap. The data from this study are quite similar to those of Humpriss et al. [25], while the greater susceptibility of females is consistent with various authors, including Levo [26].

Park et al. conducted a prospective study aimed at characterizing QoL after GKS. For this purpose, the authors used many questionnaires such as SF-36, Hearing Handicap Inventory [35], THI [36], DHI, while only a Pure Tone Audiometry (PTA) was performed as an objective evaluation. The short follow-up, and the poor compliance of an already quite small sample, greatly weakens the conclusions of the authors, according to which there was no significant decline in QoL 15–18 months after the intervention [37].

Another instrumental evaluation of the VS patient was that of Wagner et al. They enrolled 38 patients, 22 of whom microsurgically treated while 16 underwent Cyberknife treatment. Each patient was subjected to otological and neuro-ophthalmological evaluation, vestibular evaluation with Frenzel's Goggles, clinical Head Impulse Test (c-HIT) [38], Subjective Visual Vertical [39] and bithermal caloric test [40]. PTA was performed in all patients and DHI was administered to all the patients [41]. The patients were divided into two groups according to the size of the tumour (larger or smaller than 20 mm). The authors point out that the group with the smallest mass complained of more preoperative vestibular symptoms, while the loss of vestibular function was greater in the large tumours. This finding agrees with that of many other authors. The treatment method does not affect the long-term outcome of vestibular function and quality of life. We find the invitation to evaluate the impact on the quality of life of the proper choice of therapeutic timing very appropriate.

Uehara et al. studied a sample of 38 patients treated with retrosigmoid (RS) access between 2005 and 2008. The patients completed a DHI and underwent ENG with caloric irrigation and static posturography. [42] We have some doubts about the subdivision into groups (group 1, root canal paresis 0–99%, group 2, root canal paresis 100%) which could make the evaluation of the data very coarse. The small sample contributed to lowering the strength of this study. The authors also cited an under-stratification in age groups, for which we have not found evidence. Finally, a worse outcome in older patients was underlined.

Another sample of 237 patients was prospectively analysed by Breivik et al., aiming to compare the outcomes of GKS and OB management [43] Their first target was to assess the growth rate and HL secondary endpoint to assess QoL. The authors stated that both cohorts showed similar rates of vertigo and unsteadiness. This would give credit to the radiotherapy treatment for not introducing worsening symptoms. We must highlight that SF-36 and Visual Analog Scale (VAS) were used to assess the study, making the data weak on the whole.

In 2013, Batuecas et al. presented a retrospective study on 49 patients who underwent surgery for VS between 2002 and 2012 (41 RS and 8 TL surgery), who were not subjected to VRP and were observed for at least one year, using scientifically rigorous methods such as v-HIT measurement of VOR. Vestibular symptoms were assessed using the DHI questionnaire [44]. This group of authors mainly used the RS, which was feared by many due to the possible harmful effects on the vestibular apparatus caused by cerebellar retraction [45]. Authors found that patients older than 55 years have limited compensation capabilities, ending in more permanent disability after surgery. This is probably also due to

the intrinsic aging-related deterioration of vestibular performance [46–48], which agrees with numerous studies. They also clearly showed that a low level of caloric vestibular deficit is significantly related to a slower recovery. The DHI score was also higher than in patients with abnormal caloric tests, even if close to areflexia. This is consistent with Lee [49], Sughrue [50] and Tjernstrom [51]. This study is fascinating and well conducted; unfortunately, the retrospective data lack a preoperative high-rate vestibular assessment. Furthermore, the smallness of the sample helps to weaken the statistical effect. Evidence level 2b places it, in any case, among the best works on the subject.

Batuecas et al. presented another retrospective analysis of a similar sample of VS patients, underlining how the v-HIT is endowed with a greater accuracy in the study of the vestibular function compared to the caloric tests, because it is substantially a test that administers a physiological vestibular stimulus. Unlike the caloric tests which induce the vestibular stimulation through an induced motion of fluids [52], this study has a weak level of evidence. Furthermore, the authors also observed a reduced gain of VOR in the healthy side, inviting future studies to consider this type of evaluation as well [44]. The same authors further underlined that a worse preoperative deficit correlates with a faster postoperative recovery [53,54].

An interesting study is the one presented by Stavas et al., who prospectively examined a sample of 10 patients treated with SRS and examined using an objective vestibular (ENG, VNG, caloric, VEMPs) tests and DHI questionnaire [55]. They found no correlation between dose or volume constraint and vestibular function or perceived dizziness. One patient received a single fraction radiosurgery and experienced the greatest change in caloric test and DHI score. This could lead to speculation on acute dose toxicity, but a larger sample and longer follow-up is needed. Among the small sample and short-term follow-up, we found that the absence of c-VEMPs responses at baseline, due to ageing or disease, was one of the most limiting factors for further large-scale studies.

Presutti et al. examined a sample of 81 VS treated with a RS approach. Radical removal of the mass was achieved in all patients [56]. The primary endpoint of this study was facial outcome. Vestibular data are limited, but the evidence clearly shows a correlation between the absence of pre-treatment symptoms and worsening of post-operative quality of life, and the presence of preoperative vestibular imbalance correlates with a certain percentage of aggravation. The authors noted that most patients not complaining of pre-surgical vertigo reported a post-surgical worsening of QoL (57%), and few were the subjects who were already symptomatic (26%). The QoL assessment was made using generic tools such as SF-36 and Glasgow Benefit Inventory [57]. No specific vestibular questionnaires was used, and this is a major limitation, as is the lack of other objective tests.

In our review work, we also mention the article by Abboud et al., who used a small sample of 64 patients retrospectively analysed based on medical records and controlled post-operatively with Videooculography (VOG), caloric test, rotating chair, and PTA [58]. A DHI questionnaire was not used. Unfortunately, the validity of the study results is greatly undermined by the absence of objective preoperative testing.

Another prospective cohort study was conducted by Thomeer et al. on 48 patients who underwent transpetrosal surgery for VS, aiming to establish the factors influencing mid-term post-operative balance and QoL. All patients underwent VRP from the first post-operative day. Each patient was subjected to complete audiological and vestibular testing, including VEMPs and ABR, and agreed to complete a DHI questionnaire [59]. Interestingly, the tumour size and postoperative vestibular compensation was not significatively related to the self-perceived post-operative balance. This is very controversial among other authors. The authors found no age correlation with DHI scores. This is consistent with the current literature. A very interesting consideration is that patients with good preoperative vestibular compensation showed worse postoperative disequilibrium and serviceable hearing. This could be explained by the slow growth of the tumour, which would help hearing preservation, but would force a continuous preoperative central readjustment, which would suddenly fail due to surgical ablation, leading to an intense vestibular symptomatology.

This observation is consistent with those of other authors, including Enticott [27]. The authors failed to validate SVV as prognostic factor because it lacks sensitivity. Finally, the presented data failed to correlate VRP with better DHI, probably distorted by a selection bias because only heavily symptomatic patients were referred to rehabilitation programs. The final achievement of the same DHI scores of the less symptomatic cohort could be an encouragement to expand the studies on the role of VRP in these patients.

Suarez et al. used a small pilot sample to study chronic dizziness in patients undergoing VS surgery, using an assessment of gravitational vertical perception (GV) via head tilt responses (HTR) [60]. The study was designed with 6 operated patients with vestibular symptoms (not compensated) against 2 operated patients with compensated symptoms and controlled with a group of 12 healthy subjects of the same age. Patients were studied with bedside, ENG, and DHI assessments. The study was designed in a simple way, but conducted with rigor and offers some very interesting thoughts, observing how the unilateral damage is sufficient to make even the compensation mechanisms for the perception of GV not fully effective. It draws attention to the possible occurrence of subclinical damage due to brainstem and cerebellar manipulation during surgery.

Samii et al. focused on vestibular symptoms in a sample of 19 patients with disabling vestibular symptoms who underwent RS microsurgery for intracanalicular VS between 2001 and 2013 [61]. The authors randomly selected 19 intracanalicular VS patients without vestibular symptoms as the control group. The first objective of the surgery was the tumour resection: the vestibular nerve was spared as long it was not involved or not interfering with the complete mass resection. All patients in group A underwent VRP for at least one month. Most of the patients (12/19) were free from vertigo within 3 months of surgery, and this number increased within one year (17/19). The DHI score after one year was significantly reduced, and the multivariate regression showed that the preoperative DHI score affected the postoperative score within the first 3 months, while it did not affect the score after 1 year. Interestingly, the DHI score after 1 year shows no statistically significant difference when comparing sample and control group. Although the data extracted from this study are interesting, it must be recognized that the smallness of the sample and the absence of objective pre- and post-operative clinical data regarding vestibular compensation constitutes a major limitation of the study.

Hrubá et al. recently published a prospective study whose primary objective was to establish the short-term vestibular compensation capabilities of a supervised and intensive VRP program administered to a sample of 52 VS patients treated via a RS approach. During the program, 16 of them were pre-treated with IT Gentamicin, while 36 were not pre-treated [1]. The authors compared groups to evaluate the potential role of pre-treatment in speeding the compensation process. A neuro-otologic assessment with ENG, caloric test, SVV and HIT was performed in all patients, and VRP started 3 months before surgery. Statistically significant improvement was registered in SVV, posturographic parameters and ABC score following VRP in both groups. The authors conclude that the pre-treatment does not accelerate the compensation phases, arguing that the result is due to the VRP. We think that this last statement is difficult to substantiate in the absence of a control group. The dichotomy between cochlea and vestibule is interesting and worthy of further study, with dose and volume which are predictive of cochlear function and not of vestibular function.

Lee et al. presented a retrospective study of medical records of 115 patients undergoing primary GKS, collected between 2005 and 2018. In total, 37 patients developed acute vestibular syndrome within 6 months of surgery (imbalance in 23, dizziness in 11 and vertigo in 4 of them), 18 were referred to vestibular rehabilitation program (VRP), while data were recorded only for 13 [49]. The rate of symptom improvement after vestibular rehabilitation was 77% (10/13). There were no differences between subjects with and without pre-existing vestibular disorders such as migraine, benign paroxysmal positional vertigo and Meniére's disease. The authors suggest that smaller tumours have stronger correlation with developing acute vestibular symptoms compared to larger tumours. This could be due to the radiation damage to the vestibular nerve or the incomplete deafferenta-

tion, which leads to a residual nerve function. This is consistent with Sughrue [50] and Tjernstrom [51]. Lee et al. also focused on the distance from the tumour's edge to the vestibule to find correlations between the irradiated dose and the symptoms. Their data did not demonstrate a difference in this measurement between cohorts. The authors concluded that pre-existing vestibular conditions and the incidence of post-GKS vestibular side-effects have no correlation, and pre-GKS vestibular symptoms cannot predict post-GKS symptoms. While we thought it was a fascinating study, the study was limited by important methodological shortcomings, such as the absence of an objective vestibular evaluation, which appears strange for a study that aims to evaluate the impact on the vestibular apparatus. An important limitation is, once again, the lack of use of a clearly effective level test, such as the DHI, which made the anamnestic collection less effective.

Ermis et al. retrospectively collected data from 53 patients treated at a single centre between 2010 and 2016, receiving single-fraction SRS for unilateral, sporadic VS [62]. Their analyses underlined that the larger target volume (more than 6 cm$^3$), higher Koos grade (III-IV) [63], presence of pre-SRS dizziness and minimum radiation dose to the vestibule (more than $4 \pm 1.5$ Gy) were associated with patient-reported dizziness after SRS treatment. Interestingly, authors stated that patients with improved caloric function after SRS received lower mean and maximum doses. Unfortunately, this study has several limitations: data regarding vestibular symptomatology and impact on QoL were collected using a non-standardized questionnaire, and the subsample of patients with worsened caloric function after SRS has very weak statistical power, but despite the limitations, the indication to conform the irradiations so as not to exceed the 5Gy cut-off on the vestibule certainly has interesting implications on a clinical level.

Boari et al. analysed a sample of 379 patients treated with Gamma Knife Surgery (GKS), ensuring a follow-up of at least 36 months. The authors calculated the time elapsed between the onset of symptoms and the diagnosis, which was 21 months for HL, 12 months for tinnitus, and 6 months for vertigo, confirming it as the most alarming symptom for patients [64]. It is interesting to note that in this sample of patients, as many as 121 out of 379 (32%) had lived or worked in noisy environments for over 20 years, while 42 out of 379 used their mobile phone for more than 2 h a day for at least 10 years. This subpopulation also showed a median age of onset 8 years younger than moderate cell phone users (54 vs. 62) and a higher incidence of tumours on side of the ear used during phone calls. This is consistent with Hamernik [65], Edwards [66] and Preston [67], but needs a larger sample to gain statistical significance. Boari et al. observed a 2.1% and 3.2% rate of permanent vertigo and disequilibrium after GKS, respectively. They also performed a logistic regression analysis, emphasizing a higher probability for women to develop vertigo ($p = 0.012$) or imbalance ($p = 0.003$). A VS larger than 25 mm in maximum axial diameter has shown to be correlated with a higher probability of balance complications ($p = 0.041$). Tumour control was achieved in 97.1% of cases, and the morbidity rate was very low. The authors indicate that 72 patients (19% of the sample) had vertigo at the time of GKS, and 45 of these (62.5%) showed complete recovery during the follow-up. There were also 22 cases of new-onset vertigo/dizziness and 8 worsening, of which 22 relapsed during follow-up. What we can extrapolate from their study is that vertigo and dizziness post GKS, both previous and new onset, increase, then decrease until they are halved at the last check-up. The authors concluded that GKS is an effective treatment with a good safety profile and recommend it for VS diameters less than 30 mm and in young patients with well-preserved hearing. Unfortunately, this study is very focused on hearing function, leaving out a rigorous study of vestibular function, so no consistent data can be extracted.

In 2005, Hempel et al. presented a work containing the retrospective data of 125 patients who underwent GKS between 1994 and 2000, with follow-up until 2004 [68]. MRI was used for the collection of data relating to tumour control. For auditory assessment, an interview was used, and audiometry, tinnitus and vertigo were explored with a non-standardized binary questionnaire. Although this study aims to evaluate the post-GK

functional outcomes, a considerable bias in the data collection largely affects its efficacy regarding the vestibular part.

In 2019, Nilsen et al. presented a retrospective study on 433 patients aimed at studying the long-term effects of the conservative strategy on balance, dizziness and caloric function [69]. The choice of the conservative strategy was based only on physical criteria such as tumour growth (conservative if under 20 mm, GKS if 20–25 mm or smaller but growing, microsurgery if larger than 25 mm). In our opinion, it was a limitation not to evaluate the symptoms as well. Patients were administered a bithermal caloric test, a postural balance assessment, and vertigo was characterized with a VAS. The study found no significant changes in postural balance, dizziness symptoms or caloric response in the conservative arm of the sample. This is consistent with caloric tests, which seem to not deteriorate over time unless the tumour is not growing. It is rare to find studies with a follow-up period of ten years. Unfortunately, the use of a VAS scale for dizziness, the change of technology from static to dynamic posturography and the use of a caloric test rather than a more objective one are important limitations reducing the strength of this study.

An interesting point of view is the one provided by Stieglitz et al., who retrospectively analysed the intake of antiemetic drugs, subjective nausea, as well as the physical parameters of the tumour in a sample of 97 patients [70]. The data agree with those of most of the other authors regarding the greater postoperative vestibular symptoms of smaller tumours, even if the authors arrive at a different explanation, arguing a sort of habit of disequilibrium by patients with larger masses because they have been suffering for the longest time. Another peculiar standpoint is the greater consumption of antiemetics made by women, with a higher reporting rate for nausea and vomiting. The authors point out that the literature indicates a worse tolerance to pain in women [71,72] and a greater consumption of analgesics [73], imagining there is a difference in vestibular signals processing, or a possible greater susceptibility linked to hormonal influences. These are fascinating theories, as far as we currently know. We have not included the data from this study in the table, as they are extremely difficult to objectify and lacking in scientific rigour.

## 4. Discussion

Current VS management strategies show great variability among different centres; therefore, the decision-making process seems difficult to understand. The decision is generally based on tumour parameters such as the size and location of the tumour mass and the growth rate on MRI imaging. In the last decade, interest in quality of life (QoL) measurements and perspectives has increased. Handicap due to disequilibrium had the greatest negative impact on QoL. [74] It must be noted that available evidence concerning the efficacy of VS management tends to focus on removing the lesion and/or preventing growth of the tumour. Comparatively, less attention has been focused on controlling other symptoms, including persistent dizziness [75].

### 4.1. Assessment of Vestibular Deficit

Vestibular function tests such as the caloric test should be used routinely in the VS workup, using videonystagmography (VNG) or electronystagmography (ENG). These tests provide objective data about the degree of vestibular impairment and vestibular compensation; however, objective tests do not investigate the impact of the impairment on everyday life. Therefore, self-evaluation questionnaires are complementary. The DHI has proven to be a valuable and validated tool [24], capable of a more detailed assessment of functional, emotional, and physical deficits that occur secondary to balance problems or vertigo, which describes the patient's perception of vestibular symptoms. Moreover, a consensus on a universal reporting system for patients with VS was published by Kanzaki et al., in 2003, describing vestibular symptoms from a quality-of-life perspective. A grading system was proposed and included four grades: Grade I, no dizziness or disequilibrium; Grade II, occasional or slight dizziness or disequilibrium; Grade III, moderate or persistent dizziness or disequilibrium; and Grade IV, severe persistent dizziness, or disequilibrium [76]. The

caloric test stimulates the horizontal semi-circular canal, which is innervated by the SVN. The asymmetry between the two horizontal semi-circular canals is usually calculated using the Jongkees formula; a unilateral weakness less than 25% is considered normal. Some authors investigated [77] whether an ENG pattern following caloric test could be branch specific. They described pathologic caloric test findings as a marker for SVN involvement, while a tumour involving IVN showed normal caloric responses. There are also opposing opinions in the literature, such as that of Ushio et al., who found no clear evidence of correlation between caloric test, VEMPs, ABR and nerve of origin, but with tumour size [78]. There is agreement among the authors to attribute the worst response to the caloric tests to the larger masses [41,79–81].

Another vestibular test of fundamental importance is the Head Impulse Test [82]. The HIT studies the horizontal VOR and is the only bedside test that allows side-specific diagnosis in unilateral deficits. The more unilateral weakness increases, the more HIT could be effective in diagnostic pathway [83]. Video Head Impulse Test (v-HIT) increases diagnostic power by detecting even covert saccades, adding a PC-based processing of eye and head movements. When assessing a VS patient, both tests should be performed, as they elicit VOR at different rate of stimulation. The correlation between these tests is only moderate, and the caloric test shows better sensitivity than HIT. Blodow et al. [84] found a significative correlation between the caloric test impairments and tumour size and HL. The HIT findings were not statistically significant.

Some studies have demonstrated a clinical value of VEMPs in VS diagnosis, and VSs with normal hearing and negative caloric tests have been described [85]. Several authors have found absent or decreased VEMPs in at least 80% of VS, with an inversely proportional association between potential amplitude and tumour size. More medial tumours are also associated with a higher rate of VEMPs abnormalities [86,87]. The compressive effect of the tumour on the brainstem, on the spinal tract and on the nerve causes demyelination, increasing the latency of the potentials [80,88]. VEMPs complement the caloric tests in VS assessment, also filling some gaps in the latter, such as the subtle difference between hyporeflexia and areflexia [89]. They have also been described to be highly sensitive to changes in the inner ear fluid dynamics and to detect defects of the bony labyrinthine wall [90].

*4.2. Pure Tone Audiometry*

The Pure Tone Audiometry completes the triad of essential exams for a correct evaluation of VS with vestibular impairments. The type and severity of the hearing loss, as well as the monitoring of its changes, can provide useful data on tumour growth, as well as influencing treatment decisions. Using a thirty-year database, Stangerup et al. demonstrated, that 59% of patients with speech discrimination better than 70% retained good hearing after an average of 4.7 years of observation, while in the subsample with 100% discrimination, a satisfactory hearing ability at 10 years was maintained in 69% of subjects. In a subgroup of patients with small discrimination loss, 38% maintained good hearing during follow-up [91]. The same authors stated that a growth rate of less than 2.5 mm per year correlates with a higher likelihood of hearing preservation. Consequently, they propose that in patients with small tumours and normal speech discrimination, the therapeutic strategy should be chosen based on simple observation of the tumour growth rate [92]. Day et al. have postulated an association between patients with normal hearing or low-frequency deficits and small tumours, between medium or high-frequency deficits and medium-sized tumours, while in patients with pantonal and/or severe deficits, they report a correlation with tumours of dimensions greater than 2.5 cm. The mechanism of this damage would be the compression of the tumour on the cochlear nerve [93]. It is interesting to note that there is an association between small tumours and alteration of the hearing threshold at low frequencies, as is observed in cases of endolymphatic hydrops.

*4.3. Treatment Options in VS Management*

Thanks to an ever-greater early diagnostic capacity, most of the VS diagnoses occur in a mild symptomatic phase, when patients are experiencing minimal disability, adding a critical aspect in terms of management. This leads to the following question: how convenient is early intervention when the risk of collateral damage could exceed the functional deficit and associated preoperative symptoms?

Given the lack of universally recognized guidelines on therapy, the operative approach is extremely tailored to the patient, as well as highly dependent on the professional and cultural background of the attending physician.

The choice of microsurgical removal depends on several factors, such as tumour size, growth rate or severity of neuro-otological symptoms [94]. Lateral approaches to the cerebellopontine angle for VS resection include the middle cranial fossa, TL and labyrinth-sparing transmastoid approaches, such as the presigmoid/retro-labyrinthine and RS approaches [58]. The VS surgery produces a complete unilateral vestibular deafferentation, in most cases leading to acute vestibular dysfunction during the early postoperative period [24].

In patients treated with a retrosigmoid surgical approach, a higher incidence of disequilibrium is observed, probably caused by cerebellar retraction [33]. The overall prevalence of persistent postsurgical disequilibrium is ranging from 10% to 78% [95,96]. It has been observed that patients suffering from headaches who underwent microsurgery for VS show worse DHI scores than those treated with radiosurgery or observational management [97]. The surgical intervention may be associated with some other risks including HL, facial nerve palsy, brainstem/cerebellar injury, cerebrospinal fluid fistula, hydrocephalus, meningitis, as well as general risks associated with microsurgery and anaesthesia [41].

Sometimes, the unaffected vestibular nerve ends up flattened on the surface of the tumour. In these cases, nerve sectioning is chosen to allow safe removal of the tumour and to avoid excessive manipulation which could endanger the cochlear branch of the eighth cranial nerve, as well as the facial nerve. The resection of the mass and the opening of the canal relieve the pressure on the healthy vestibular nerve. It is likely to hypothesize a scenario in which the positive effect of the surgery benefits both the deafferentation of the diseased vestibular nerve and the decompression of the healthy one. As a result, the unstable vestibular impulses to the vestibular nuclei will be stabilized. Disabling vestibular symptoms should be considered an indication for surgery, even in otherwise asymptomatic patients with intracanalicular VS [61].

Regarding the therapy of cases of VS with vertigo, the clinician has very little objective to refer to in addition to the clinical parameters. Patients' opinion regarding the incidence or severity of vertigo, as well as the impact of the disease on quality of life, is relied upon when planning therapeutic intervention.

The first-line treatment for small-to-moderate VS is considered by many authors to be sterotaxic radiosurgery (SRS) [98]. Despite this role of primary importance recognized by an increasing number of authors, the impact of this technique on the perception of vertigo depending on the applied dose has only been marginally studied [62].

SRS for VS involves the single-session application of a high single dose of radiation to a precisely controlled intracranial location, aiming to inhibit further tumour growth by inducing a vascular necrosis [68]. Usually, hospitalization is not required. Some possible complications include injury of the facial or cochlear nerve, vertigo, hydrocephalus, headaches, and tinnitus [38].

During the last decade, FRT has been increasingly used as the first choice RT method in VS patients. Dose fractionation allows the differential sparing of normal tissues. Some preliminary studies over hearing and facial preservation showed high success rates [99]. A retrospective comparison with SRT showed better hearing preservation in the FRT group [100]. Most of the studies are weakened by short follow-up. A longer study showed mid-term outcomes less promising than those observed in preliminary studies [29]. The failure of RT is a controversial diagnosis. Most of the VS are subject to mass swelling for the first few months after irradiation, then they can reduce in size or remain consistent [101].

When a continuous volumetric increase is shown during a 3-year-follow-up, some authors refer to it as treatment failure [102]. In these cases, it is preferable to closely observe how the neoplasm behaves, if no new functional limitation or symptomatology arises, waiting for the danger to subside; otherwise, a more invasive approach may be justified.

### 4.4. "Wait and Scan" Observational Option

Conservative management of VS with regular MRI has become a common strategy in recent years. With the increased availability of MRI, this "wait and scan" policy has become feasible, since about 50% of small- to medium-sized tumours do not grow when observed for 5 years [69], while a definitive treatment is required for larger VS which must be managed with a surgical approach or with radiation therapy.

If the tumour shows evident growth and causes an increase in symptoms or the onset of new symptoms, it is necessary to evaluate the abandonment of the wait and scan for operative strategies.

Vertigo has been reported to be one of the risk factors for the growth of VS [103]. The impact of vertigo attacks on perceived QoL appears to be far greater than for other symptoms, including permanent instability. In some patients, vertigo attacks are unresponsive to medical treatment, causing substantial physical and social constraints. However, vertigo as an indication for intracanalicular VS surgery has been occasionally mentioned, as it is rarely the only presenting symptom in VS patients [32]. Most "wait and scan" management studies have a short follow-up window; this makes data weak. Most authors showed that the incidence and rate of growth reduces across time. There is also evidence of tumour growth after a long quiescence [104]. Consequently, it may be questioned what interval is needed between follow-up assessments, but the need to continue scanning throughout the patient's life is unquestionable. It must be noted that recent evidence suggests poorer long-term outcomes following conservative management [105].

Following the wider use of "wait and scan" strategy, the European Academy of Otology and Neuro-Otology (EAONO) published a position statement based on an extensive review of the existing literature [106]. The EAONO proposed a protocol to closely scan rapid-growing VS first, then a life-long follow-up to detect late growth. The protocol suggests, after the initial diagnosis, a new MRI assessment after 6 months, then annually for 5 years, then two more in the next 4 years and then one every 5 years. These are a lot of MRI scans over time, and as a result, the risk of nephrogenic fibrosis increases due to the precipitation of gadolinium [107]. This is a critical issue for observational strategy. Non-contrast MRI, such as constructive interference or 3D-FIESTA, has proven to be effective in follow-up protocols for VS [108,109]. Finally, we also observed that the authors often disagree when defining tumour dimensions: some are using the measurement of the greatest diameter, while others calculate the volume on the two major axes. Thus, they define the growth rate differently.

### 4.5. The Role of Vestibular Compensation (VC)

The sudden unilateral vestibular deafferentation generates a strong imbalance in the discharge of the vestibular nerves of both sides: the spontaneous ipsilesional firing rate drops and the inhibitory drive increases from the contralesional side, making the total imbalance worse [110]. The VC takes place through restoration, habituation and adaptation processes. Restoration does not take place, as the nerve is damaged or sectioned; thus, all effort must be focused on habituation and adaptation processes, (new operating strategies and suppression of faulty responses). The VRP needs to be performed early and actively, during the time window of the vestibular system's plastic reorganization [111].

The process of central compensation and recalibration starts after each acute vestibular asymmetry, being effective in weeks to months; in this scenario, exercises of vestibular rehabilitation can play a major role [112], reducing spontaneous nystagmus and improving posture. Typical rehabilitation programs involve assisted eye movement and postural exercises [113]. In VS, it is well known that throughout the period of tumour growth, an

ongoing vestibular compensation process helps the subject to maintain balance and to reduce oscillopsia. Nevertheless, after surgical operation or SRS, an important reduction of vestibular inputs will take place [44]. It is not easy to predict which level of vestibular functionality will be achieved for each patient after VS treatment. Typically, elderly subjects or people with additional central nervous disorders face problems compensating for the acute vestibular loss following treatment. In cases of simultaneous acute vestibular loss and cerebellar lesions (e.g., due to the surgery), the compensatory process may be prolonged or incomplete [111]. In VS patients, another way to improve the compensation process is vestibular prehabilitation that is performed before surgical treatment [114]. It consists of vestibular exercises to induce motor training and to optimize the function of vestibular and postural reflexes. Subsequently, vestibular function on the pathological side is ablated with intratympanic gentamicin while the rehabilitation program continues. Finally, surgery is performed when total or near total loss of vestibular function and a compensated vestibulo-ocular reflex are achieved.

## 5. Limitations of the Study

The purpose of this article is to present an overview of the main pitfalls of the therapeutic management of VS with vestibular symptoms, based on a selection of articles available in the literature.

It is necessary to draw attention to the numerous limitations of this work, including having restricted the research field only to works dealing with the vestibular aspect, the small size of our sample (which was not selected according to the criteria of a universal and systematic review) and the fact that the selected articles are often tainted by further biases such as small samples and often selected on the basis of the type of technique being studied. This is a particularly limiting aspect: most of the studies examined presented homogeneous samples of patients with similar clinical characteristics. Generally, MS samples are composed of larger or more symptomatic or aggressive tumours for which there is often no reasonable therapeutic alternative to compare, while the samples examined in the OB studies were generally affected by small and less symptomatic tumours. In studies of SRS or FRT methods, the tumours were often smaller, and one of the main objectives was the saving of auditory function. In comparative studies, we have often seen subdivision into groups with very different symptoms and clinics. The difficulty of comparing such different studies with each other in the hope of obtaining a certain scientific reliability is obvious. Therefore, even the comparison of the treatment success rates specific to the modalities reported by the various authors is difficult task. It is difficult to draw coherent conclusions by cross-examining the results of these studies.

Our work aims to offer the reader a concise view of recent evidence, and not to act as a comparative test between techniques which, for the above reasons, are difficult to compare. In this scenario, the need to share standardized diagnostic framework schemes, universally recognized indices and create large prospective studies, comparisons and, finally, randomized controlled trials to compare techniques and outcomes appears mandatory. Only this will overcome current low-level evidence, such as expert opinion.

## 6. Conclusions

Balance preservation represents an essential objective in virtuous management of VS. To investigate this issue, we reviewed the most recent literature analysing the vestibular function's level in patients who underwent surgery or radiotherapy for VS treatment. We immediately noticed extensive use of generic or custom-made questionnaires, rather than validated vertigo/dizziness assessment, which made most of the studies unreliable and difficult to compare. Moreover, an objective assessment of vestibular signs and symptoms was lacking in most of the studies reviewed, reducing the validity of the results.

We do not consider it acceptable to analyse the QoL of VS patients without objectively testing their vestibular function and without administering the DHI questionnaire. Even if we accept only a strict questionnaires-based assessment, it would be useful to associate

a psychological-level test, given that the perception of QoL also varies according to the patient's psychological state. The cochleo-vestibular tests are fast and largely available at low cost; therefore, they should be mandatory when assessing patients with VS.

Despite the commitment of many authors to retrospectively retrieve data from their case studies, each centre has its own preference regarding the therapeutic approach; this influences the selection criteria of the sample and makes comparison with other studies proposing alternatives management options difficult.

What is striking is the limited follow-up in most of the studies. It is likely that no author can predict what the long-term outcome will be, regardless of the therapeutic strategy used!

The natural history of VS growth is elusive: the tumour may grow continuously or to a certain size, followed by inactivity or even shrinkage. A cautious and observational approach in mild symptomatic/asymptomatic and small tumours is preferred. MRI intervals should be narrow within the first 5 years to intercept disease recurrence and calculate growth rate; thereafter, they may widen, but not stop. If active treatment is needed, intervention in the early stage of disease seems to have best results and less complication with RT, but the data are weak. Microsurgery is often the preserver of larger, more symptomatic lesions or chosen based on surgeon or patient preference, in most cases allowing complete removal of the tumour.

Prehabilitation seems to be promising; VRP has a role in speeding and enhancing the natural VC processes.

The need for multicentre, randomized, and controlled studies that explore all current therapeutic strategies is strong and unavoidable. A scientifically shared guideline is needed to overcome the approximation of third-class evidence.

**Author Contributions:** Conceptualization, D.P., F.M.G. and G.C.; methodology, D.P., A.A. and P.V.; writing—original draft preparation, D.P.; data curation, A.S., M.R. and F.R.; writing—review and editing, G.C. and F.M.G. All authors have read and agreed to the published version of the manuscript.

**Funding:** This research received no external funding.

**Institutional Review Board Statement:** Not applicable.

**Informed Consent Statement:** Not applicable.

**Data Availability Statement:** Not applicable.

**Acknowledgments:** We thank Giovanna Bitonti and Chiara Covelli for their assistance throughout all aspects of our study.

**Conflicts of Interest:** The authors declare no conflict of interest.

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
