# Peer review of "Vestibular Impairment in Patients with Vestibular Schwannoma: A Journey through the Pitfalls of Current Literature"

_audiolres, doi:10.3390/audiolres13020025_

Round 1

Reviewer 1 Report

This manuscript is a review of many articles on vestibular impairment in VS patients and provides a detailed description of the diagnoses and treatments. However, we think that the following points need to be revised.

Major points

1) The text is too long and very difficult to thoroughly read. Especially in the results section, each paper is presented in detail, but it is only a summary of each and does not indicate the main points. It would be better to have a subtitle and categorize them like in the discussion section.

2) In the results section, it should be clearly stated how many papers were reviewed and through what process the papers were finally selected.

Minor points

1) I think there are too many line breaks.

2) There are mixed descriptions of QoL and QOL

3) In the discussion section, the description of hearing is included in the Assessment of vestibular deficit section, but it should be listed separately.

Reviewer 2 Report

The authors detail the relationship between the surgical strategy of vestibular schwannoma and impaired vestibular function and vestibular rehabilitation. Vestibular symptoms after vestibular schwannoma are summarized and quantitative assessment. Strategic options are provided for clinicians in considering the compensation of vestibular function in surgical patients as well as in rehabilitation. The authors' study is of important clinical value.

Author Response

Dear auditor,

Thank you for your precious considerations.
I hope that our manuscript can help those who will deal with the subject.

The authors

Reviewer 3 Report

Dear Authors,
Thank you for your hard effort to extrapolate data from the scientific literature
concerning the vestibular evaluation in case of vestibular schwannoma.
The great number of reports on this topic and the non-comparable data,
inevitably made the Results and the Discussion quite fragmented, but this is
understandable given the topic.
The current strong debate on the diagnostic and therapeutic management
options of vestibular schwannoma makes your review work of great interest
and relevance. I think you should emphasize more to the Readers how an adequate
pre- and post-operative vestibular testing could facilitate
vestibular rehabilitation in case of vertigo/dizziness following treatment.
I really appreciated your honest description of the strong limitations of your
study, but the purpose of your manuscript is clear, in order to raise awareness
on the important topic regarding the management of vestibular impairment in patients with vestibular schwannoma.
The Conclusion is focused on the topic of the manuscript and the
References are more than complete in my opinion.
For these reasons I consider your manuscript suitable for publication. Thanks again.

Author Response

(The authors gave the same response as above.)

Round 2

Reviewer 1 Report

It is still very long and requires a great deal of effort to read, but if it cannot be shortened by any means, it must be tolerated.